# Protocol for a phase 3 trial to evaluate the effectiveness and safety of a heterologous, two-dose vaccine for Ebola virus disease in the Democratic Republic of the Congo

Deborah Watson-Jones [1,2] Hugo Kavunga-Membo,[3] Rebecca F Grais,[4] Steve Ahuka,[3] Natalie Roberts,[5] W John Edmunds,[1] Edward M Choi [1] Chrissy H Roberts,[1] Tansy Edwards [1] Anton Camacho,[4] Shelley Lees,[1] Maarten Leyssen,[6] Bart Spiessens,[6] Kerstin Luhn,[6] Macaya Douoguih,[6] Richard Hatchett,[7] Daniel G Bausch,[1,8] Jean-Jacques Muyembe,[9] DRC-EB-001 protocol writing team

For numbered affiliations see end of article.

**Correspondence to**
Dr Deborah Watson-Jones;
deborah.Watson-Jones@lshtm.ac.uk

## ABSTRACT

**Introduction** Ebola virus disease (EVD) continues to be a significant public health problem in sub-Saharan Africa, especially in the Democratic Republic of the Congo (DRC). Large-scale vaccination during outbreaks may reduce virus transmission. We established a large population-based clinical trial of a heterologous, two-dose prophylactic vaccine during an outbreak in eastern DRC to determine vaccine effectiveness.

**Methods and analysis** This open-label, non-randomised, population-based trial enrolled eligible adults and children aged 1 year and above. Participants were offered the two-dose candidate EVD vaccine regimen VAC52150 (Ad26.ZEBOV, Modified Vaccinia Ankara (MVA)-BN-Filo), with the doses being given 56 days apart. After vaccination, serious adverse events (SAEs) were passively recorded until 1 month post dose 2. 1000 safety subset participants were telephoned at 1 month post dose 2 to collect SAEs. 500 pregnancy subset participants were contacted to collect SAEs at D7 and D21 post dose 1 and at D7, 1 month, 3 months and 6 months post dose 2, unless delivery was before these time points. The first 100 infants born to these women were given a clinical examination 3 months post delivery. Due to COVID-19 and temporary suspension of dose 2 vaccinations, at least 50 paediatric and 50 adult participants were enrolled into an immunogenicity subset to examine immune responses following a delayed second dose. Samples collected predose 2 and at 21 days post dose 2 will be tested using the Ebola viruses glycoprotein Filovirus Animal Non-Clinical Group ELISA. For qualitative research, in-depth interviews and focus group discussions were being conducted with participants or parents/care providers of paediatric participants.

**Ethics and dissemination** Approved by Comité National d'Ethique et de la Santé du Ministère de la santé de RDC, Comité d'Ethique de l'Ecole de Santé Publique de l'Université de Kinshasa, the LSHTM Ethics Committee and the MSF Ethics Review Board. Findings will be presented

## Strengths and limitations of this study

► This is the first large-scale trial designed to measure effectiveness of a two-dose Ebola vaccine through open label delivery to general populations in an Ebola-affected area.

► This is the first clinical trial to administer this vaccine regimen to pregnant women.

► The pragmatic design facilitated trial delivery during an active Ebola epidemic where the future location of new cases is uncertain.

► A test-negative case-control analysis to estimate vaccine effectiveness in a situation where a randomised controlled trial with a control arm was not possible.

► A non-randomised design is a limitation of this study.

to stakeholders and conferences. Study data will be made available for open access.

**Trial registration number** NCT04152486.

## INTRODUCTION
### Background and rationale

Ebola virus disease (EVD) is caused by Ebola viruses (EBOV) and has a case fatality rate ranging from 30% to 90%.[1] Bats are thought to be the reservoir, with human infection resulting from contact with excrement or saliva from bats or other intermediate animal hosts.[2–5] Large outbreaks are fueled almost exclusively by subsequent human-to-human transmission.[2 6 7]

Vaccination is now a proven preventative measure for EVD control. During the 2013-2016 EVD outbreak in West Africa, a single dose of the rVSV-ZEBOV-GP vaccine (ERVEBO, Merck & Co.) given through a

phase 3 cluster-randomised, ring-vaccination trial in Guinea showed 100% protective efficacy against EVD between day 10 and day 31 post randomisation.[8] In 2016, the WHO Strategic Advisory Group of Experts on Immunization recommended the rapid deployment of rVSV-ZEBOV-GP for EVD outbreaks under an expanded access (compassionate use) protocol and the vaccine was licensed by the Democratic Republic of the Congo (DRC), Burundi, Ghana and Zambia in February 2020.[9 10]

Since the discovery of EBOV in 1976, the DRC has recorded 12 outbreaks. On 1 August 2018, the 10th EVD outbreak was declared in North Kivu Province.[11] The outbreak progressively expanded across a large, highly populated area, characterised in recent years by civil unrest, armed militia groups, internally displaced persons and a substantial United Nations peacekeeping mission.[12] The outbreak response focused on strategies that were previously adopted for the control of EVD outbreaks but was periodically hindered by attacks on Ebola Treatment Centres (ETCs) and on response workers. New cases continued to emerge in North Kivu and neighbouring provinces, prompting consideration of additional vaccine programmes that might assist in preventing the spread of this infection to unaffected communities. In early 2019, WHO conducted an evaluation of the suitability of candidate EVD vaccines for further clinical study and recommended several vaccines including VAC52150 (Ad26.ZEBOV, MVA-BN-Filo), a heterologous two-dose prophylactic vaccine manufactured by Janssen Vaccines and Prevention B.V.[13]

## Objectives

Following the WHO evaluation, we are conducting an open-label, non-randomised, population-based trial in North Kivu Province. This trial is named 'Evaluation of a heterologous, two-dose preventive Ebola vaccine for effectiveness and safety in the Democratic Republic of the Congo'. The protocol number is DRC-EB-001. The primary objective is to estimate the vaccine effectiveness (VE) of Ad26.ZEBOV, Modified Vaccinia Ankara (MVA)-BN-Filo for prevention of EVD in adults and children aged 1 year or above. Secondary objectives are to: (1) assess the safety of Ad26.ZEBOV, MVA-BN-Filo, (2) estimate the coverage of dose 1 and dose 2 of Ad26.ZEBOV, MVA-BN-Filo overall and in different target groups (by gender, age and location) and (3) to explore the knowledge and perceptions of persons eligible for large-scale delivery of a preventative EVD vaccine with a two-dose vaccine strategy. An exploratory objective is assessing immunogenicity of Ad26.ZEBOV, MVA-BN-Filo in a subset of participants who received dose 2 outside the recommended interval between doses 1 and 2. This is the first study providing Ad26.ZEBOV, MVA-BN-Filo to pregnant women (table 1).

## METHODS
### The vaccine

The vaccination regimen comprised two vaccine candidates, both given as a 0.5 mL intramuscular injection in the upper section of the deltoid muscle or in the thigh in the case of young children:

1. Ad26.ZEBOV: a monovalent vaccine expressing the full-length glycoprotein (GP) from EBOV Mayinga, that is produced in the human cell line PER.C6, $5 \times 10^{10}$ viral particles and given at day zero.
2. MVA-mBN226B, or MVA-BN-Filo: a multivalent vaccine expressing the GP of EBOV (100% homologous with the GP expressed by Ad26.ZEBOV), Sudan and Marburg Musoke viruses; and the nucleoprotein of Taï Forest virus, $1 \times 10^8$ infectious units given at day 56 (−14 day+28 day).

This regimen has been evaluated for immunogenicity and safety in 11 clinical trials in the UK, USA and East and West Africa, including previously EVD-affected countries.[14–21] When administered in a 0, 56 day schedule in phase 2 and 3 studies, geometric mean concentrations of immunoglobulin G (IgG) binding antibody to the EBOV GP measured by the Filovirus Animal Non-Clinical Group (FANG) ELISA 21 days post dose-2 were 3810–11 790 EU/mL in healthy participants. The FANG ELISA assay has been validated for use in human sera.[22] Unblinded safety data from 2390 adults showed only mild-to-moderate adverse events of short duration with no sequelae.[17] No safety concerns were raised in HIV-infected individuals.[18] In 649 children, the vaccine was highly immunogenic and had a similar safety profile as adults, with no Suspected Unexpected Serious Adverse Reactions.[19] The European Commission granted marketing authorisation for Ad26.ZEBOV, MVA-BN-Filo on 1 July 2020.[20 21]

For this study, vaccine was donated by Janssen Vaccines and Prevention B.V. Vials were stored and shipped according to Good Manufacturing Practice guidelines with maintenance of the cold chain from Belgium to the vaccine depot in Goma, DRC and onto the vaccination clinics.

### Trial design

The protocol development team and investigators considered various trial designs that would allow estimation of vaccine efficacy or effectiveness in the context of an epidemic and vaccination activities with the rVSV-ZEBOV-GP vaccine. The value of placebo-controlled randomised controlled trial (RCT) designs was recognised but, given the ongoing EVD outbreak at the time of initiation of the study and taking into account sociocultural sensitivities and civil instability in the region, an RCT was not considered feasible or ethical. Rather, we adopted a test-negative case-control design, in which VE is estimated by comparing the history of vaccination in cases (test positives) compared with that in controls (test negatives), among individuals presenting with suspected EVD, with cases and controls defined retrospectively after EVD test results were confirmed. The test-negative design allows evaluation of VE in the general population without randomisation, and has been proposed as a valid approach for evaluation of EVD vaccines when RCTs are not feasible.[23 24] The validity of this approach has been

**Table 1** The WHO Trial Registration Data Set for the DRC-EB-001 trial

| Data category | Information |
|---|---|
| Primary registry and trial identifying number | ClinicalTrials.gov NCT04152486 |
| Date of registration in primary registry | 5 November 2019 |
| Source(s) of monetary or material support | Coalition for Epidemic Preparedness Innovations (CEPI), the Paul G. Allen Family Foundation, the UK Department for International Development (DFID) and Wellcome |
| Sponsor | London School of Hygiene & Tropical Medicine (LSHTM) |
| Sponsor contact and contact for public queries | Deborah Watson-Jones deborah.watson-jones@lshtm.ac.uk |
| Public title | Effectiveness and safety of a Heterologous, Two-dose Ebola Vaccine in the DRC |
| Scientific title | Evaluation of a heterologous, two-dose preventive Ebola vaccine for effectiveness and safety in the Democratic Republic of the Congo |
| Countries of recruitment | The Democratic Republic of the Congo |
| Health condition(s) studied | Ebola virus disease (EVD) |
| Intervention(s) | Ad26.ZEBOV, MVA-BN-Filo |
| Key inclusion criteria | Healthy volunteers<br>Aged ≥1 year<br>Both sexes (including pregnant and breastfeeding women) |
| Key exclusion criteria | Known EVD infection<br>Having received any experimental EVD vaccines within 1 month |
| Study type | Interventional |
| Allocation | Open label, single arm |
| Primary purpose | Prevention |
| Phase | III |
| Date of first enrolment | 14 November 2019 |
| Target sample size | 500 000 |
| Recruitment status | Complete |
| Primary outcome(s) | To estimate the vaccine effectiveness of population-based vaccination delivery with the Ad26.ZEBOV, MVA-BN-Filo vaccine regimen for the prevention of EVD in adults and children aged 1 year or above. (timeframe: from 2 weeks after intervention to end of study) |
| Key secondary outcomes | To assess safety of the Ad26.ZEBOV, MVA-BN-Filo vaccine (timeframe: From dose 1 vaccination to 1 month post dose 2)<br>To estimate the coverage of dose 1 and dose 2 of the Ad26.ZEBOV, MVA-BN-Filo vaccine regimen overall and in different target groups (timeframe: vaccination period)<br>To explore the knowledge and perceptions of persons eligible for large-scale delivery of a preventative Ebola vaccine with a two-dose vaccine strategy (timeframe: vaccination period) |
| Other outcomes | To assess immunogenicity of the Ad26.ZEBOV, MVA-BN-Filo vaccine in a subset of participants who received dose 2 outside the recommended interval between dose 1 and dose 2. |
| Ethics review 1 | Approved by LSHTM Research Ethics Committee<br>Date of approval: 15 October 2019<br>Name and contact details: Rebecca Carter ethics@lshtm.ac.uk |
| Ethics review 2 | Approved by Médecins Sans Frontières Ethics Review Board<br>Date of approval: 15 October 2019<br>Name and contact details: Raffaella Ravinetto raffaella.ravinetto@gmail.com |
| Ethics review 3 | Approved by Comité d'éthique de l'école de santé publique, Université de Kinshasa<br>Date of approval: 22 October 2019<br>Name and contact details: Willy Bongopasi bongopasi@gmail.com |
| Ethics review 4 | Approved by Comité national d'éthique et de la santé, Ministère de la Santé Publique, République Démocratique du Congo (RDC)<br>Date of approval: 23 October 2019<br>Name and contact details: Félicien Munday feli1munday@yahoo.fr |

Ad26, Adenovirus serotype 26; EVD, Ebola virus disease; MVA, Modified Vaccinia Ankara.

demonstrated by comparing results from test-negative case-control studies with those from randomised clinical trials of influenza and rotavirus vaccines. The two type of study yielded similar VE estimates.[25 26]

### Trial setting

This trial was planned to start in November 2019 and to end in February 2022. After discussion with local authorities, we initiated the trial in two administrative health areas in Goma, the provincial capital of North Kivu Province (population 634 197 in 2020). The choice of these locations was determined based on their potential risk for EVD transmission since there were strong ethnic, family and business links between these two health areas and the epidemic epicentres of Beni and Butembo, with frequent travel between the communities. Provincial administrative and health authorities provided data on the number of residents of these areas as well as the number of individuals working in health and community health settings. These people were provided with information about when and where vaccination would take place. We planned to subsequently move to health areas closer to the outbreak epicentre, guided by data on estimated risk of EVD, population density, security issues and transport links from Goma. However, plans to expand into new study sites were abandoned when the outbreak came under control and was declared over on 25 June 2020. As a consequence there was no possibility of acquiring VE data through the trial as the epidemic had not spread to the initial trial areas.

### Sample size

The required sample size for the primary effectiveness evaluation was 110 cases (test positives), to provide 80% power with alpha=0.05 for a two-sided test to demonstrate 70% VE under assumptions for coverage (percentage of vaccinated test negative individuals) ranging from 20% to 70% and ratio ranging from 1 to 4 test negatives per positive case (table 2).

This sample size also provides 80% power to demonstrate 90% VE under assumptions for coverage ranging from 10% to 70% and ratio of test positives to test negatives ranging from 1 to 4 test negatives to a test positive case. It will also provide 80% power to demonstrate 50% VE under assumptions for coverage ranging from 50% to 70% and ratio of test positives to test negatives ranging from 2 to 4 test negatives to a test positive case.

We estimated that vaccination of approximately 500 000 people with Ad26.ZEBOV, MVA-BN-Filo in transmission areas would achieve the sample size of 110 confirmed EVD cases under the following assumptions: (1) an EVD incidence of 4.1/10 000 in the absence of Ad26.ZEBOV/MVA-BN-Filo-induced immunity, corresponding to the median attack rate per health area after the first year of the epidemic in the DRC; and (2) 30% of confirmed cases had received the rVSV-ZEBOV-GP (in accordance with surveillance data) and would therefore be excluded from the effectiveness analysis. We assumed a homogeneous

**Table 2** Sample sizes for a test-negative case-control study design, assuming 80% power to detect an OR=1-VE, by per cent coverage among controls (those testing negative) and number of controls per case, with a two-sided alpha=0.05 test.

| Vaccine efficacy (VE; %) | Coverage (% vaccinated among controls) | Sample size (no. of cases) by number of controls per case | | | |
|---|---|---|---|---|---|
| | | 1 | 2 | 3 | 4 |
| 90 | 5 | 210 | 171 | 157 | 149 |
| | 10 | 103 | 84 | 77 | 73 |
| | 20 | 50 | 40 | 37 | 35 |
| | 30 | 32 | 26 | 23 | 22 |
| | 50 | 19 | 15 | 13 | 12 |
| | 70 | 14 | 11 | 9 | 9 |
| 70 | 5 | 418 | 333 | 303 | 288 |
| | 10 | 211 | 167 | 152 | 144 |
| | 20 | 107 | 84 | 76 | 72 |
| | 30 | 74 | 57 | 52 | 49 |
| | 50 | 50 | 38 | 34 | 32 |
| | 70 | 46 | 34 | 30 | 28 |
| 50 | 5 | 962 | 749 | 677 | 641 |
| | 10 | 492 | 383 | 345 | 326 |
| | 20 | 260 | 201 | 181 | 171 |
| | 30 | 186 | 143 | 128 | 120 |
| | 50 | 137 | 103 | 92 | 86 |
| | 70 | 141 | 105 | 92 | 86 |
| 30 | 5 | 3070 | 2350 | 2109 | 1989 |
| | 10 | 1593 | 1218 | 1092 | 1029 |
| | 20 | 866 | 659 | 590 | 555 |
| | 30 | 638 | 483 | 432 | 406 |
| | 50 | 499 | 375 | 334 | 313 |
| | 70 | 553 | 412 | 364 | 340 |

attack rate, while recognising that it could vary according to the location of vaccine deployment and the evolution of the epidemic.

After the VE analysis had been conducted based on 110 cases, the study team planned discussions to review the findings and the value of continuing to enrol participants.

Two subsets of individuals, 1000 vaccinated individuals (500 adults and 500 children) and 500 pregnant participants (250 pregnant predose 1 and 250 pregnant within 30 days of dose 2), were enrolled for more active follow-up for safety reporting (table 3). The sample size allowed for 95% probability to detect at least one vaccine related serious adverse event (SAE) if the true incidence of that SAE is at least 1.2% with 250 individuals, 0.6% with 500 individuals, or 0.3% with 1000 individuals.

### Community engagement and feedback

Prior to recruiting participants, we set up a specific community engagement structure. Its mission was to

**Table 3** Time and event schedule for intervention delivery and safety evaluations

| Preparation | Up to 1 month before study starts | D0 Dose 1 visit | D7 (−3/+7 day) | D21 (−6/+7 day) | D56/ Dose 2 visit (−14/+28 day) | D7 post dose 2 (−3/+7 day) | D21 post dose 2 (−3/+7 day) | 1 month post dose 2 (−7 day/+1 month) | 3 months post dose 2 (±14 day) | 6 months post dose 2 (−14/+28 day) | 3 months post delivery |
|---|---|---|---|---|---|---|---|---|---|---|---|
| Mapping and selection of study sites | X | | | | | | | | | | |
| Community engagement and social mobilisation | X | | | | | | | | | | |
| Consent, vaccination dose 1 | | X | | | | | | | | | |
| Vaccination dose 2 | | | | | X* | | | | | | |
| SAE assessment | | | | X† | X† | | | X‡ | | | |
| Safety subset | | | | | X | | | X§ | | | |
| All pregnant women | | | | | X¶ | | | X | | | X** |
| Pregnancy subset | | X | X†† | X†† | X | X†† | | X†† | X†† | X†† | X‡‡ |
| Immunogenicity subset | | | | | X§§ | | X§§ | | | | |

*Dose 2 may be given between D42 and D84. Participants who return for dose 2 after 84 days will still be vaccinated.

†D56 safety data are collected at dose 2 visits.

‡Passive serious adverse event (SAE) recording from main study participants through participant-initiated phone calls or other contact with team, until 1 month post dose 2.

§Safety subset: 1000 participants will be actively followed up for SAEs by phone at 1 month post dose 2.

¶If study vaccinations have been paused, all pregnant participants who have not received dose 2 will be called at D56 (±7 days) to collect safety data.

**All pregnant women will have their birth outcomes collected around 3 months post delivery (between 1 and 3 months). When feasible, team to visit if the woman is not contactable by telephone.

††Pregnancy subset: safety telephone calls to collect SAEs at D7 and D21 post dose 1, and then at D7, 1 month, 3 months and 6 months post dose 2, unless delivery is before these time points.

‡‡100 infants born to women in the pregnancy subset will be given a full clinical examination around 3 months after delivery (between 1 and 6 months).

§§Immunogenicity subset: a subset of paediatric and adult participants will have a venous blood draw at the dose 2 visit before vaccination. A second blood sample will be taken 21 days (−3/+7 day) after their dose 2 vaccination. Enrolment of this subset will continue until at least 50 adults and 50 children have had blood samples collected for immunogenicity at dose 2 and 21 days post dose 2.

## Box 1 DRC-EB-001 inclusion and exclusion criteria for vaccination

### Inclusion criteria

1. Must provide a written or witnessed (if illiterate) informed consent form indicating that he or she understands the reasons for the study and is willing to participate in the study and to be vaccinated. If less than 18 years old, must have a parent or guardian that is able to meet this criterion.
2. Must be aged 1 year or older.
3. Must be healthy in the investigator's clinical judgement as assessed on the day of vaccination.
4. Must be willing to have a photograph taken.
5. Must be available and willing to participate for duration of study visits and follow-up.

### Exclusion criteria

1. Known history of Ebola virus disease (EVD).
2. Has received any experimental EVD vaccine less than 1 month prior to visit 1.
3. Known allergy or history of anaphylaxis or other serious adverse reactions to vaccines or vaccine products, egg and egg proteins or gentamicin.
4. Presence of an acute illness (excluding minor illnesses such as mild diarrhoea or mild upper respiratory tract infection) or temperature ≥38.0°C at visit 1 (dose 1 visit). Participants with such symptoms will be temporarily excluded from vaccination at that time but may be rescheduled for vaccination at a later date if feasible (and if within 84 days of the first dose).
5. Presence of significant conditions or clinically significant findings at the vaccination visit for which, in the opinion of the investigator, vaccination would not be in the best interest of the participant.
6. History of recurrent generalised hives.

Note: Participants who have received treatment for acute, uncomplicated malaria are eligible for vaccination if at least 3 days have elapsed from the conclusion of a standard, recommended course of therapy for malaria. Rescheduling for dose 1 or dose 2 is possible in case of acute illness at the time of planned vaccination as long as rescheduling for dose 2 is within 84 days of dose 1.

inform, sensitise and engage directly with the local population. Initially overseen by Médecins Sans Frontières France (MSF), World Vision was involved in the introduction of specific techniques, including mobile messaging.

Within each health area, Community Advisory Committees were constituted. They conducted initial meetings with local political and administrative leaders, social groups and organisations, local associations and other non-governmental organisations, and traditional and religious leaders, and their input was included in the planning of the trial implementation. We then conducted regular, separate discussion meetings with these different groups every 2 weeks to receive and integrate feedback on social harms, individual and community level risks, perceptions about the vaccine and the study, legal or administrative complaints and vaccination implementation issues.

### Eligibility, recruitment and enrolment

Using predefined eligibility criteria (box 1), we invited all adults and children aged 1 year or greater, including pregnant and breast feeding women, to participate if they lived or worked in the selected health zones and, at the time of vaccination, planned to remain there for 1 month post dose 2. Health workers were specifically encouraged to participate. We provided information about the study to local populations through the distribution of posters and flyers, banner displays in the target health areas, radio and television broadcasts and interviews during church gatherings and through interactive street visits using loudspeakers. Until the outbreak was declared over, we provided participants with information on where to present for care if they developed EVD symptoms and informed participants that they could still receive rVSV-ZEBOV-GP as recommended by WHO if eligible.[27] We instructed Ebola Emergency Treatment Centres (ETCs) on how to report that a presenting patient had received Ad26.ZEBOV, MVA-BN-Filo. We required a parent or caregiver to accompany any child aged less than 18 years to the study vaccination site.

### Intervention delivery

We established six study vaccination sites within the participating health zones in Goma. Communicating in Swahili, French or a local dialect, Nande, we conducted symptom screening and a temperature check for possible EVD for participants at a triage desk before they could proceed into the clinic. We provided a short medical consultation to participants presenting at triage with fever and/or illness and including, if appropriate, a test for malaria, and treatment if positive, as well as free treatment for other minor illnesses according to national medical protocols. A postvaccination medical consultation was also offered for any participants complaining of illness during the observation period. We then referred participants to the nearest appropriate health facility if any further medical intervention was required.

At the first visit, delegated medical staff of the trial confirmed eligibility, explained the study aims and procedures, answered any questions, and sought informed consent (and informed assent for children aged 12–17 years), with witnessed consent for illiterate individuals. We offered urine pregnancy tests at the time of vaccination if a participant believed that she might be pregnant and/or if her last menstrual period was more than 1 month ago. We gave participants the Ad26.ZEBOV vaccination and asked them to remain at the site for 15 min to monitor for any immediate adverse effects. We provided participants with a laminated vaccination card with their photograph, which we asked them to present at subsequent visits. Enrolment and dose 1 vaccination took place between 14 November 2019 and 29 February 2020.

We collected data on place of residence and telephone numbers from all trial participants and contacted them through mobile phone calls and automated short message service (SMS) to remind them to attend for dose 2 and followed up, with home visits if these were needed. At the clinic visits for dose 2, similar triage, eligibility and health checks were conducted. Participants who attended

more than 84 days post dose 1 were still able to receive the second dose until the study vaccination sites closed on 10 February 2021.

After the study start, all individuals presenting to an ETC, peripheral health facility or an EVD testing centre were expected to have their rVSV-ZEBOV-GP and Ad26. ZEBOV, MVA-BN-Filo vaccination status confirmed as part of routine data collection, with this information being recorded in the routine Ministry of Health EVD surveillance database. This included recording the participant identification (ID) number from the study vaccination card. If the patients were not able to present their cards at the ETC but mentioned being a study participant, ETC staff requested the family to bring the card to the ETC. For lost cards, we instructed ETC staff to collect the patient's name, vaccination and clinic site details, and, if possible, the last vaccination date, and provide this information to the study data management team to retrieve the participant's ID number and manually complete the ETC line-list.

Due to the waning EVD outbreak, the Trial Steering Committee decided to suspend dose 1 vaccination on 29 February 2020, after 20 427 participants had been vaccinated. Dose 2 vaccination continued, recognising that, while the objective of evaluating VE could no longer be met, the other objectives were still achievable. However, due to the COVID-19 outbreak in DRC, all study vaccinations were suspended from 10 April to 14 September 2020, while special precautions to ensure the safety of participants and staff from COVID-19 were established. As a result of suspension of activities, the protocol was amended (10 August 2020, version 7.0) to include immunogenicity assessment of the impact of delayed dose 2 vaccination beyond 84 days post dose 1 and safety follow-up windows (table 3). Following implementation of infection control measures, vaccination continued and dose 2 delivery was completed on 9 February 2021.

In addition to the main study, we enrolled participants into three subsets, with consent sought in separate sections in the main study informed consent and assent forms (online supplemental appendixs 1–10):

1. Immunogenicity substudy: 57 non-pregnant adults and 90 children (54 children aged 4–11 years and 36 adolescents aged 12–17 years) were enrolled in an immunogenicity substudy. At selected vaccination centres, we collected venous blood samples (5 mL per visit for adults and children aged 6–17 years and 2.5 mL per visit for children aged 4–5 years) from consenting participants at the dose 2 visit and 21 days later (window −3/+7 days). We made 0.5 mL serum aliquots, which were stored at −20°C in the L'Institut National de Recherche Biomédicale (INRB) laboratory in Goma. One aliquot was shipped to Q2 Solutions in the USA for measurement of IgG binding antibodies to EBOV GP by FANG ELISA. The remaining aliquots are being stored in Goma until the end of study and will then be destroyed. We provided participants with an immunogenicity subset attendance card with a photograph and ID number for identification at the 21 days post dose 2 visit. We deferred blood drawing if participants were unwell at the second visit. Enrolment and site visits of this subset were completed on 7 December 2020. Details of laboratory handling of biological specimens can be found in the Laboratory Analytical Plan.

2. Safety subset: 1000 participants (approximately 500 adults and 500 children predose 1) were asked to consent to a telephone call at 1 month post dose 2 to collect data on SAEs.

3. Pregnancy subset: up to 500 pregnant participants (250 pregnant predose 1 and 250 pregnant within 30 days of dose 2) were asked to consent to receive telephone calls to collect SAEs at D7 and D21 post dose 1, and at D7, 1 month, 3 months and 6 months post dose 2 (unless delivery has occurred before these time points) and data are being collected on pregnancy outcomes as described below.

## Safety measurements

Following each vaccination, we gave participants instructions to contact the study team for any SAEs occurring up to 1 month post dose 2 and where to seek care for medical problems, including antenatal care for pregnant women (table 3). Whenever possible, we documented the diagnosis or syndrome related to SAEs, rather than multiple symptoms. Investigators then assessed any potential causal relationship to vaccination. We followed up SAEs to resolution or stabilisation, irrespective of severity or whether considered vaccine related. We allowed unscheduled study follow-up visits on adverse events at the investigator's discretion.

Follow-up of pregnant women is ongoing. We compiled a registry of pregnant women and we encouraged women who become pregnant within 1 month of any vaccine dose to contact the team. We are telephoning all known pregnant women 1–3 months after delivery to record birth outcomes or, when possible, a study clinician visits them in person if they are not contactable by telephone. In the pregnancy subset, we actively follow by telephone 250 women who were pregnant at the time of dose 1 and up to 250 who became pregnant within 30 days (1 month) of administration of either dose 1 or 2. We make calls to collect SAE data at D7 (−3/+7 day) and D21 (−6/+7 day) post dose 1, D7 (−3/+7 day) post dose 2, and at 1 month (−7 day/+1 month), 3 months (±14 day) and 6 months (−14/+28 day) post dose 2 (unless delivery occurs before these time points). We are performing a clinical examination on the first 100 infants born to women in the pregnancy subset at around 3 months of age (ranging from 1 to 6 months). We offer a small bag of rice, cassava or equivalent to mothers returning to the clinic for infant safety follow-up visits.

In the safety subset, we actively followed the first 500 adults and first 500 children enrolled by telephone 1 month (−7 days/+1 month) after administration of dose 2.

For acute medical problems, we provide care to participants with referral to health clinic and hospitals as necessary. The trial sponsor and funder have put in place global and local clinical trial insurance covers to compensate any potential harm to participants.

## Social science methods

We are using multiple modalities to explore the socioeconomic and sociocultural contexts of EVD and the study, including perception of illness and disease, trust in medical research and the motives for the study, key community dynamics and power relations, barriers and opportunities. This information can be used to trace any ongoing rumours and concerns emerging at the community level and to provide appropriate feedback to the study team.

The social science component of the study includes the following:
1. 30 in-depth interviews: 5 women, 5 men, 5 girls and 5 boys aged 12–17 years old and 10 parents/care providers of children aged 1–11 years, post dose-1 or 2.
2. Eight focus group discussions: including adult participants, parents of children aged 1–11 years, and boys and girls aged 12–17 years.
3. Up to 10 key informant interviews: local and national stakeholders.
4. Participant exit interviews at vaccination visits to document immediate concerns related to clinic experiences.
5. Ethnographic observation in study clinics and key sites (eg, markets, motorbike parking grounds).

At the beginning of all interviews, the social science team ask participants to state that they consent to taking part, recording this both digitally and by hand on the consent form, or only by hand if permission is not given for digital recording. Parents are being asked to provide consent for children under 17 years and children aged 12–17 years to provide assent. Interviews are being conducted in the participant's/parent's/guardian's primary language. Exit interviews are extremely basic and rapid and simply record whether participants had any immediate concerns they wished to report. If in-person interviews are not possible, we conduct phone interviews with adult participants, with researchers reading out information from an information sheet.

Interviews are conducted in French, Swahili or Nande and then translated to English. We remove personal identifiable information prior to uploading recordings to a SharePoint which is only accessible to the study team. We give each interview participant a unique ID number and delete their phone numbers from study databases immediately after use.

## OUTCOMES

The primary study outcomes are:

1) The numbers and odds of vaccination status in EVD cases and in EVD-negative controls in a test-negative case-control study with a target sample size of 110 laboratory confirmed EVD cases matched to controls who test negative for EVD. Effectiveness is derived from the OR for vaccination in cases compared with controls to calculate VE.

The secondary outcomes include:
1. The number and proportion of adults and children with solicited and unsolicited SAEs from the date of first vaccination to 1 month post dose 2. Data on SAEs within 1 month post dose 2 that are considered related to vaccination with Ad26.ZEBOV, MVA-BN-Filo in adults and children.
2. The number and proportion of adults and children receiving dose 1.
3. The number and proportion of adults and children receiving dose 2.
4. The number of participants participating in in-depth interviews and focus group discussions on participant and community perceptions of the trial and on vaccine acceptability.

The exploratory outcome includes:
1. The level of IgG antibodies against EBOV GP in samples collected from participants in the immunogenicity subset who received dose 2 outside the recommended interval between dose 1 and dose 2.

## DATA COLLECTION, MANAGEMENT AND ANALYSIS
### Data collection

Data collection is ongoing. We collect data using electronic case record forms (eCRFs) developed using Open Data Kit (ODK) software on password protected tablets, and on standardised paper CRFs when eCRF use is not possible.[28] We designed a modular system of ODK forms in English and French to collect data on different aspects of the study, including consent, vaccinations and errors on eCRF completion. Prior to starting the study, we provided and reviewed guidelines for CRF completion with study site personnel. Automated programmes in R or STATA summarise study progress and create line-lists for follow-up and dose 2 visits, and to identify errors. Data can be exported as delimited text files. Automated checks ensure data completion. Study team members receive daily updates and reports via a password protected dashboard. eCRFs are also transferred into REDCap Good Clinical Practice (GCP) compliant software for an audit trail of data cleaning. Data collected in paper-based CRFs are double-entered directly into a REDCap database.

After encryption, data are sent via end-to-end encrypted https protocols to a server hosted at the London School of Hygiene & Tropical Medicine (LSHTM), with ongoing transfer of data to servers hosted by INRB and Epicentre via SSH File Transfer Protocol. Only the data management team at Epicentre and INRB are able to decrypt the raw data. Study personnel save, store, back-up, transfer and share data in encrypted file formats and over encrypted transmission routes. To the extent possible, we use deidentified data for operational and analytic purposes.

Permissions to decrypt data are limited to key personnel. Working copies of databases such as those used by staff who are responsible for progress monitoring and statistical analysis are automatically stripped of personal identification data via a process applied by data managers at Epicentre.

We anonymise interview transcripts and field notes and download them to password protected and encrypted computers for transcription and translation and analysis using NVIVO software. We keep all recordings, transcripts and field notes in locked filing cabinets at the study's Goma data centre, before moving them to the long-term archive at INRB Kinshasa. Details of database validation, data collection, coding, security, automatic, query, cleaning and safety data reconciliation can be found in the Data Management Plan.

## VE analysis

We define cases (test-positives) as laboratory-confirmed EVD cases as per DRC Ministry of Health centralised reporting and controls (test-negatives) as those with suspected EVD but whose laboratory tests were negative. Prior to the end of the outbreak, we envisaged that identification of controls would be retrospective, blind to vaccination status, on individuals residing or working in areas that were offered vaccination prior to presentation with symptoms at health facilities, with geographical matching to allow cases and controls to have had equal opportunities to have been vaccinated. Frequency matching or adjustments using regression techniques for other characteristics such as age, sex and possibly also exposure to EBOV were to be described in the Statistical Analysis Plan.

Since the likely ratio of test-negatives to test-positives at screening centres was unknown, it was recognised that, if a high number of individuals with a very low percentage of confirmed cases presented in a short period of time, random sampling of test-negatives on a frequency matched basis may be required.

If the duration of Ebola outbreak had permitted measurement of VE, individuals who received rVSV-ZEBOV-GP, or the first but not the second dose of Ad26.ZEBOV, MVA-BN-Filo or who received the second dose less than 21 days prior to testing for EVD would have been excluded from the primary analysis.

For the primary analysis, the odds of having been fully vaccinated (having received the two doses, in the right order, at least 1 month apart, at least 21 days before the onset of symptoms) would have been compared with the odds of not being vaccinated between the cases and controls through the OR, providing an estimate of VE: (VE (%)=(1−OR)×100), with regression analyses allowing for appropriate account of any matching and adjustment for potential confounders.

We also planned to examine VE of at least one dose of the vaccine and development of disease within 21 days after the second dose as secondary analyses.

## Safety analysis

Safety analyses will include all participants receiving at least dose 1. We will summarise the incidence of SAEs overall, by time since vaccination and by age and whether or not vaccination was received during pregnancy. The incidence of SAEs will also be tabulated by MedDRA preferred term and system organ class (Medical Dictionary for Regulatory Activities version 23.1).

## Immunogenicity subset analysis

We will determine the number of overall participants with detectable antibodies and maximum titres post vaccination, including analysis by age group and by time since dose 1 vaccination. We will make descriptive comparisons with other studies of the same vaccine in sub-Saharan Africa where dose 2 was given according to the recommended 56-day interval and, for adults, with other delayed dose 2 immunogenicity data. We will analyse continuous immunological parameters (eg, geometric mean concentration of IgG binding antibodies to EBOV GP) and responder rates, and assess the impact of extending the interval between the two doses.

More detailed information on analysis populations, data handling and subgroup analyses is provided in the statistical analysis plan.

## Qualitative research analysis

We are conducting inductive and deductive analysis of qualitative data collected by in-depth interviews, focus group discussions and exit interviews. All recordings are being transcribed into French and imported to NVIVO V.12. Data are being coded by constant comparative methods to create emergent themes.

## Trial management

LSHTM is the study sponsor. A Trial Steering Committee holds regular meetings by teleconference to monitor study progress (online supplemental appendix 11). An independent Data Safety and Monitoring Board (DSMB) periodically reviews the safety data and meets according to the participant recruitment timeline specified in the DSMB Charter (online supplemental appendix 11). Fatal, life-threatening events and Suspected Unexpected Serious Adverse Reactions are reported to the DSMB within 48 hours and other SAEs within 7 days. An independent GCP monitor checks to compliance with the protocol, GCP, standard operating procedures and informed consent processes through review of documents for 5% of study participants through on-site visits and remote monitoring. There will also be one on-site GCP audit by a sponsor representative not involved in conducting the trial before primary data collection has completed. Please see online supplemental appendix 12 for the responsibilities of the study sponsor and funders.

## ETHICS AND DISSEMINATION

The protocol was reviewed and approved by the Comité National d'Ethique et de la Santé and Comité d'Ethique

de l'Ecole de Santé Publique in DRC, the LSHTM Ethics Committee and the Ethics Review Board of MSF (table 1). This study is being implemented according to the Declaration of Helsinki principles and the principles of the International Conference on Harmonisation (ICH) Guideline for GCP. Since the original approved protocol version 5.0, dated 12 September 2019, there has been two protocol amendments. This article is based on protocol version 7.0, dated 10 August 2020. All versions and amendments have been approved by the four ethics committees listed above and by the DRC Direction de la Pharmacie et du Médicament. The study has been registered in Clinical-Trials.gov (ClinicalTrials.gov Identifier: NCT04152486).

The population of North Kivu Province is considered vulnerable. Study participation is strictly voluntary. Participants can withdraw from the trial at any time. If provided, the reasons for withdrawal will be recorded. Both communities and, at the time of the outbreak, individuals were informed that study participation did not mean that other EVD infection protection and control measures should be discarded and that they should endeavour to attend for the second dose.

Results of the study will be presented to relevant stakeholders in DRC, shared with WHO's technical expert groups and published in open access journals. Any research articles generating from this trial will be subject to the Publication Policy and the Authorship Guidelines agreed by all partners.

### Data availability statement
The rights of study subjects and partners, the sharing of data between partners and the transfer of data to external third party will be governed by the Data Sharing Agreement. Deidentified participant-level data collected in this trial will be disseminated through a FAIR-compliant data repository, such as the LSHTM Data Compass (https://datacompass.lshtm.ac.uk/), from 6 to 60 months after the publication of the main trial results. Other study documents (eg, full protocol, statistical codes, Statistical Analytical Plan, DSMB Charter) will be available on request to Deborah Watson-Jones (corresponding author, ORCID: 0000-0001-6247-1746), Tansy Edwards (study statistician, ORCID: 0000-0002-6110-014X) or Edward Choi (study coordinator, ORCID: 0000-0002-8148-120X).

### Patient and public involvement statement
Because we invited the general population to participate in this trial, no patients were involved in the discussions on trial design, conduct, reporting or dissemination. Prior to trial commencement, we engaged the Goma population through consultation with local authorities and leaders. Refer to the main article for details.

**Author affiliations**
[1]London School of Hygiene & Tropical Medicine, London, UK
[2]Mwanza Intervention Trials Unit, Mwanza, Tanzania, United Republic of
[3]L'Institut National de Recherche Biomédicale, Goma, Democratic Republic of the Congo
[4]Epicentre, Paris, Île-de-France, France
[5]MSF, Paris, France
[6]Janssen Vaccines and Prevention BV, Leiden, Zuid-Holland, The Netherlands
[7]Coalition for Epidemic Preparedness Innovations, Oslo, Norway
[8]UK Public Health Rapid Support Team, Public Health England and LSHTM, London, UK
[9]L'Institut National de Recherche Biomédicale, Kinshasa, Democratic Republic of the Congo

**Acknowledgements** We thank participants, medical and nursing staff for their participation in this study and World Vision for community engagement support.

**Collaborators** London School of Hygiene & Tropical Medicine (LSHTM):Deborah Watson-Jones, W John Edmunds, Edward M Choi, Chrissy H Roberts, Shelley Lees, Tansy Edwards, Daniela Manno, Peter G Smith, Brian Greenwood, Daniel G Bausch. L'Institut National de Recherche Biomédicale (INRB) : Jean-Jacques Muyembe, Steve Ahuka, Hugo Kavunga-Membo. Epicentre: Rebecca Grais, Susan Rattigan, Anton Camacho, Grace Mambula. Médecins Sans Frontières, France (MSF): Natalie Roberts, John Johnson, Patient Mumbere Kighoma, Marie Burton. Janssen Vaccines and Prevention B.V.: Maarten Leyssen, Macaya Douoguih, Bart Spiessens. Coalition for Epidemic Preparedness Innovations (CEPI): Richard Hatchett, Nathalie Imbault, Gerald Voss, Melanie Saville. Inserm-IRD-Université de Montpellier: Eric Delaporte. World Health Organization: Ira M Longini.

**Contributors** DWJ, JJM, DGB, CHR, RG, WJE, NR, PGS, TE, HKM, ML, AC, EMC, BG, IML, ED and SA conceived and designed the study. DWJ and EMC wrote the protocol with inputs from RG, SR, DGB, HKM, PGS, NI, TE, CHR, WJE, BG, BS, KL, MD, RH, SL, AC and the protocol writing team. TE and PGS provided statistical advice. DWJ and EMC wrote the article. RG, DGB, HKM, PGS, NI, TE, CHR, WJE, BG, JJ, BS, KL, ML, MD, NR, RH, AC, SR, SL, SA and JJM reviewed and revised the manuscript. All authors and the protocol writing team approved the final submitted manuscript.

**Funding** This study is supported through funds from the Coalition for Epidemic Preparedness Innovations (CEPI) [FELS1903] and the Paul G. Allen Family Foundation. This work was also supported by the UK Department for International Development (DFID) and Wellcome [220506/Z/20/Z] and by the European Union's Horizon 2020 research and innovation programme under grant agreement No 857935. This publication reflects only the authors' view and the European Commission is not responsible for any use that may be made of the information it contains. CEPI, Wellcome and DFID advised on the study design and protocol.

**Competing interests** DWJ, BG and LSHTM are partners on two research consortia (EBOVAC1, EBOVAC3) with Janssen Vaccines and Prevention B.V. funded by the European Commission.

**Patient consent for publication** Consent obtained directly from patient(s)

**Provenance and peer review** Not commissioned; externally peer reviewed.

**ORCID iDs**
Deborah Watson-Jones http://orcid.org/0000-0001-6247-1746
Edward M Choi http://orcid.org/0000-0002-8148-120X
Tansy Edwards http://orcid.org/0000-0002-6110-014X

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
