## [Reviewer comments · BMJ Open]

ARTICLE DETAILS

TITLE (PROVISIONAL)	Protocol for a phase 3 trial to evaluate the effectiveness and safety of a heterologous, two-dose vaccine for Ebola virus disease in the Democratic Republic of the Congo
AUTHORS	Watson-Jones, Deborah; Kavunga-Membo, Hugo; Grais, RF; Ahuka, Steve; Roberts, Natalie; Edmunds, John; Choi, Edward; Roberts, Chrissy; Edwards, Tansy; Camacho, Anton; Lees, Shelley; Leyssen, Maarten; Spiessens, Bart; Luhn, Kerstin; Douoguih, Macaya; Hatchett, Richard; Bausch, Daniel; Muyembe, Jean-Jacques

VERSION 1 – REVIEW

REVIEWER	Choi, Mary J. National Center for Emerging and Zoonotic Infectious Diseases
REVIEW RETURNED	13-Sep-2021

GENERAL COMMENTS	The authors faced several challenges to protocol implementation. Despite this, the data that can be gleaned (immunogenicity, SAEs, effect of lengthening the time between vaccine doses) are still of value and will contribute to the body of knowledge of this vaccine.
---

REVIEWER	Sanyaolu, Adekunle Federal Ministry of Health
REVIEW RETURNED	23-Sep-2021

GENERAL COMMENTS	This Protocol for a phase 3 trial to evaluate effectiveness and safety of a heterologous, two-dose vaccine for Ebola virus disease in the Democratic Republic of the Congo is well written and well designed considering the limitation in the study area. EVD is a deadly disease that continues to be a significant public health problem in affected countries; hence vaccination is a very important control measure for the disease. There are few typo errors that require correction by the authors. Add 'the' before 'effectiveness' in the title "Protocol for a phase 3 trial to evaluate the effectiveness and safety of a heterologous, two-dose vaccine for Ebola virus disease in the Democratic Republic of the Congo" On page 17, line 32, rewrite the statement for clarity i.e...may have been required ormay be required.
---

VERSION 1 – AUTHOR RESPONSE

Thank you to the reviewers and editors for their comments on our manuscript entitled "Protocol for a phase 3 trial to evaluate effectiveness and safety of a heterologous, two-dose vaccine for Ebola virus disease in the Democratic Republic of the Congo".

We have amended the following sections of the manuscript and the entry on [clinicaltrials.gov](https://www.clinicaltrials.gov) (<https://www.clinicaltrials.gov/ct2/show/study/NCT04152486>), as per the BMJ recommendations.

BMJ: Please revise the 'Strengths and limitations' section of your manuscript (after the abstract). This section should contain up to five short bullet points, no longer than one sentence each, that relate specifically to the methods only.

We have amended the 'Strengths and limitations' section to five bullet points, a single sentence each, all relating to the trial design.

BMJ: Please ensure that your protocol reports all outcome measures for your trial and ensure that the primary and secondary outcome measures are consistent between your protocol article and the trial registry.

The study outcomes in the manuscript and the clinical trial registry have been amended to exactly match the outcome measures in the protocol as follows.

Vaccine safety ("the number and proportion of adults and children with solicited and unsolicited serious adverse events") has been moved to secondary outcomes.

The social science outcome below has been added to the manuscript as a secondary outcome.

"The number of participants participating in in-depth interviews and focus group discussions on participant and community perceptions of the trial and on vaccine acceptability."

The Immunogenicity outcome below has been added to the manuscript as an exploratory outcome.

"The level of IgG antibodies against EBOV GP in samples collected from participants in the immunogenicity subset who received dose 2 outside the recommended interval between dose 1 and dose 2."

The following pregnancy safety outcome measure has been deleted from the [clinicaltrials.gov](https://www.clinicaltrials.gov) registry to match the protocol.

"Number and proportion of pregnant participants with solicited and unsolicited serious adverse events including congenital abnormalities in their infants."

BMJ: Please ensure that the information provided in your protocol article is consistent with that included in the trial registry. For example, the sample size. Please update the manuscript and/or trial registry accordingly.

We have updated the trial information on [clinicaltrials.gov](https://www.clinicaltrials.gov) as follows.

"Safety will be assessed in a safety subset of 1000 individuals and a pregnancy subset of up to 500 women will be followed to delivery. The first 100 infants born to these pregnant participants will be given a clinical examination at 3 months post-delivery.

The target sample size for the primary effectiveness evaluation is 110 laboratory-confirmed EVD cases."

An exploratory objective is to assess the immune response at before the second dose and 21 days after the second dose (MVN-BN-Filo) in a subgroup of 50 adults and 50 children who receive dose 2 beyond the recommended 56-day interval.

BMJ: Please include the planned start and end dates for the study in the methods section.

The start date (November 2019) and the end date (February 2022) have been added to the 'methods' section of the manuscript, under 'Trial Setting'.

BMJ: Funding Information: You have indicated a funder/s for your paper. Please ensure to provide an award/grant number for your funder/s in the main document file and in ScholarOne.

We have added the CEPI grant number "FELS1903" to the manuscript and in ScholarOne.

BMJ reviewer: There are few typo errors that require correction by the authors.

Thank you for pointing these out. The typos have now been corrected.